# ROBUSTNESS TO CORRUPTION IN PRE-TRAINED BAYESIAN NEURAL NETWORKS

**Xi Wang**
College of Information and Computer Science
University of Massachusetts Amherst
xwang3@cs.umass.edu

**Laurence Aitchison**
Department of Computer Science
University of Bristol
laurence.aitchison@bristol.ac.uk

## ABSTRACT

We develop ShiftMatch[1], a new training-data-dependent likelihood for robustness to corruption in Bayesian neural networks (BNNs). ShiftMatch is inspired by the training-data-dependent "EmpCov" priors from Izmailov et al. (2021a), and efficiently matches test-time spatial correlations to those at training time. Critically, ShiftMatch is designed to leave the neural network's training time likelihood unchanged, allowing it to use publicly available samples from pre-trained BNNs. Using pre-trained HMC samples, ShiftMatch gives strong performance improvements on CIFAR-10-C, outperforms EmpCov priors (though ShiftMatch uses extra information from a minibatch of corrupted test points), and is perhaps the first Bayesian method capable of convincingly outperforming plain deep ensembles.

## 1 INTRODUCTION

Neural networks are increasingly being deployed in real-world, safety-critical settings such as self-driving cars (Bojarski et al., 2016) and medical imaging (Esteva et al., 2017). Accurate uncertainty estimation in these settings is critical, and a common approach is to use Bayesian neural networks (BNNs) to reason explicitly about uncertainty in the weights (MacKay, 1992; Neal, 2012; Graves, 2011; Blundell et al., 2015; Gal & Ghahramani, 2016; Maddox et al., 2019; Aitchison, 2020a;b; Ober & Aitchison, 2021; Unlu & Aitchison, 2021; Khan & Rue, 2021; Daxberger et al., 2021). BNNs are indeed highly effective at improving uncertainty estimation in the in-distribution setting, where the train and test distributions are equal (Zhang et al., 2019; Izmailov et al., 2021b). Critically, we also need to continue to perform effectively (or at least degrade gracefully) when presented with corrupted inputs. Superficially, BNNs seem like a good choice for this setting: we would hope they would give more uncertainty in regions far from the training data, and thus degrade gracefully as inputs become gradually more corrupted, and thus diverge from the training data.

However, recent work has highlighted that BNNs including with gold-standard Hamiltonian Monte Carlo (HMC) inference can fail to generalise to corrupted images, potentially performing worse than ensembles (Lakshminarayanan et al., 2017; Ovadia et al., 2019; Izmailov et al., 2021a;b). Izmailov et al. (2021a) gave a key intuition as to why this failure might occur. In particular, consider directions in input space with zero variance under the training data. As the weights in this direction have little or no effect on the output, any weight regularisation reduces the weights in these directions to zero. The zero weights imply that these directions continue to have no effect on the outputs, even if corruption subsequently increases variance in these input directions. However, BNNs do not work like this. BNNs sample weights in these zero-input-variance directions from the prior. That is fine in the training data domain, as there is no variance in the input in those directions, so the non-zero weights do not affect the outputs. However, if corruption subsequently increases input variance in those directions, then those new high-variance inputs will interact with the non-zero weights to corrupt the output.

Izmailov et al. (2021a) suggested an approach for fixing this issue, by modifying the prior over weights at the input layer to reduce the variance in the prior over weights in directions where the inputs have little variance. While their approach did outperform BNNs with standard Gaussian priors, it performed comparably to deep ensembles (Izmailov et al., 2021a, their Figs. 4,11,12). This failure is surprising: part of the promise of BNNs is that they should perform well for corrupted inputs by giving more uncertainty away from the training data.

---

[1]Code available at https://github.com/xidulu/ShiftMatch

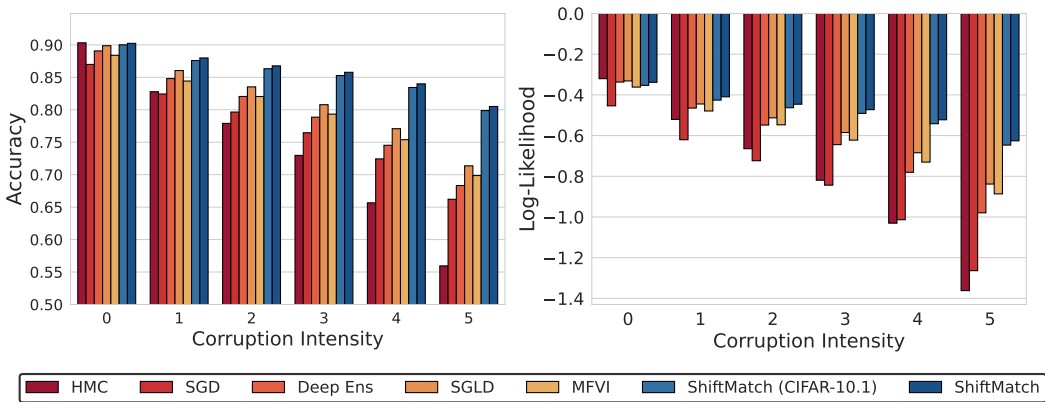

Figure 1: Performance of HMC samples ,SGD, Deep Ensemble, SGLD, mean-field VI extracted from (Izmailov et al., 2021b) and ShiftMatch (applied on HMC samples from Izmailov et al., 2021b) on CIFAR-10-C averaged over 16 different corruption types with 5 intensity level (1 to 5). ShiftMatch (CIFAR-10.1 Recht et al., 2019) uses the 2000 new images from CIFAR-10.1 to compute the training statistics, demonstrating that we get only a small performance penalty if the full training data is not available, but we can get access to data from roughly the same distribution. Intensity 0 stands for clean CIFAR-10 test set without corruption. ShiftMatch significantly improves the robustness against corruption compared with plain HMC and even outperforms Deep Ensemble, but uses additional information from a minibatch of corrupted test datapoints, which is not used by the other methods.

However, the performance of any Bayesian method on in-distribution and corrupted inputs in depends heavily on the choice of model (prior and likelihood combined). We begin by noting that the training-data-dependent EmpCov priors from Izmailov et al. (2021a) can be equivalently viewed as training-data-dependent likelihoods. We then develop a new training-data-dependent likelihood, ShiftMatch, which has two advantages over EmpCov priors (Izmailov et al., 2021a): First, EmpCov priors apply only to the input layer, so might not be effective for more complex corruptions which are best understood and corrected at later layers. In contrast, our likelihoods modify the activity at every layer in the network, so have the potential to fix complex, nonlinear corruptions. Second, EmpCov modifies the prior over weights, preventing the use of publically available samples from BNNs with standard Gaussian priors (Izmailov et al., 2021b), which is especially important as some gold-standard Bayesian sampling methods are extremely expensive (e.g. by Hamiltonian Monte Carlo, HMC in Izmailov et al. (2021b) took one hour to get a sample on a ResNet trained on CIFAR-10 using "a cluster of 512 TPUs" Izmailov et al., 2021b). In contrast, ShiftMatch keeps the prior and training-time likelihood unchanged, allowing us to directly re-use gold-standard HMC samples from Izmailov et al. (2021b). Indeed, ShiftMatch is highly efficient as it does not require further retraining or fine-tuning at test time, allowing us to e.g. use a very large batch size.

We found that ShiftMatch improved considerably over all previous baselines for BNN robustness to corruption, including BNNs with standard Gaussian and EmpCov priors, and non-Bayesian methods like plain deep ensembles (Fig. 1,3). Further, ShiftMatch can be combined with non-Bayesian methods: we found significant improvement for all methods tested: stochastic gradient descent (SGD), ensembles, and Bayes (HMC) (Fig. 4), and ShiftMatch can be scaled to ImageNet (Fig. 5), where it offers improved performance over test-time batchnorm (Nado et al., 2020).

## 2 BACKGROUND

In BNNs, we use Bayes theorem to compute a posterior distribution over the weights $\mathbf{w}$ given the training input, $\mathbf{X}_{\text{train}}$, and labels, $\mathbf{y}_{\text{train}}$,

$$P\left(\mathbf{w}|\mathbf{X}_{\text{train}}, \mathbf{y}_{\text{train}}\right) = \frac{P\left(\mathbf{y}_{\text{train}}|\mathbf{X}_{\text{train}}, \mathbf{w}\right) P\left(\mathbf{w}\right)}{P\left(\mathbf{y}_{\text{train}}|\mathbf{X}_{\text{train}}\right)}. \tag{1}$$

At test-time, we use Bayesian model averaging (BMA) to compute a predictive distribution over test labels, $\mathbf{y}_{\text{test}}$, given test inputs, $\mathbf{X}_{\text{test}}$, by integrating over the posterior,

$$\mathrm{P}\left(\mathbf{y}_{\text{test}} \mid \mathbf{X}_{\text{test}}, \mathbf{X}_{\text{train}}, \mathbf{y}_{\text{train}}\right) = \int \mathrm{P}\left(\mathbf{y}_{\text{test}} \mid \mathbf{X}_{\text{test}}, \mathbf{w}\right) \mathrm{P}\left(\mathbf{w} \mid \mathbf{X}_{\text{train}}, \mathbf{y}_{\text{train}}\right) d\mathbf{w}. \tag{2}$$

In practice, directly using Bayes theorem is intractable in all but very low dimensional settings. As such, alternative approaches, ranging from (HMC) (Neal et al., 2011; Izmailov et al., 2021b), variational inference (Graves, 2011; Blundell et al., 2015), stochastic gradient Langevin dynamics (SGLD) (Welling & Teh, 2011; Zhang et al., 2019) or expectation propagation (EP) (Hernández-Lobato & Adams, 2015) are usually used. HMC is usually regarded as the "gold-standard", as it will produce exact samples given enough time to sample, while the other methods here will not.

## 3 RELATED WORK

The closest prior work is Izmailov et al. (2021a), which introduces the "EmpCov" prior that uses the training inputs, $\mathbf{X}_{\text{train}} \in \mathbb{R}^{N_{\text{train}} \times M}$, to specify the prior over weights,

$$\mathrm{P}_{\text{EmpCov}}\left(\mathbf{w} \mid \mathbf{X}_{\text{train}}\right) = \mathcal{N}\left(\mathbf{w}; \mathbf{0}, \tfrac{1}{M}\left(\tfrac{1}{N_{\text{train}}}\mathbf{X}_{\text{train}}^{\top}\mathbf{X}_{\text{train}}\right)\right) \tag{3}$$

Here, $N_{\text{train}}$ is total number of training examples, $M$ is the number of features in the input layer, and $\mathbf{w} \in \mathbb{R}^{M}$ is a weight vector mapping from the inputs to a single feature in the first hidden layer. The covariance of the Gaussian prior over weights, $\frac{1}{MN_{\text{train}}}\mathbf{X}_{\text{train}}^{\top}\mathbf{X}_{\text{train}}$, is proportional to the empirical covariance of the training inputs. Thus, the prior suppresses weights along directions with low input variance. Interestingly, the idea of using the inputs to specify a prior over weights arises earlier, in the Zellner G-prior (Zellner, 1986). Remarkably, Izmailov et al. (2021a) and Zellner (1986) propose priors that are in some sense "opposite": while the EmpCov prior suppresses weights in directions with low input variance, the Zellner-G prior magnifies weights in those directions, so as to ensure that all directions in input space have the same impact on the outputs,

$$\mathrm{P}_{\text{Zellner-G}}\left(\mathbf{w} \mid \mathbf{X}_{\text{train}}\right) = \mathcal{N}\left(\mathbf{w}; \mathbf{0}, \tfrac{1}{M}\left(\tfrac{1}{N_{\text{train}}}\mathbf{X}_{\text{train}}^{\top}\mathbf{X}_{\text{train}}\right)^{-1}\right). \tag{4}$$

Of course, the best prior depends on the setting: we suspect that the Zellner-G prior is most appropriate when informative features have very different scales, whereas the EmpCov prior is most appropriate for e.g. images, where the low-variance input directions are likely to be uninformative.

One approach to using unlabelled data to improve robustness to corruption is to keep batchnorm on at test-time, so the batch statistics are computed on corrupted test data (Li et al., 2016; Carlucci et al., 2017; Nado et al., 2020; Schneider et al., 2020; Benz et al., 2021). There is considerable confusion about whether batchnorm can be applied to BNNs, with recent papers arguing that batchnorm cannot be formulated as a likelihood that factorises across datapoints (Izmailov et al., 2021b; D'Angelo & Fortuin, 2021; Noci et al., 2021). However, we argue that valid likelihoods do not need to factorise across datatpoints. We introduce such a likelihood for BNNs, which behaves like applying batchnorm during training and keeping batchnorm on at test-time. Surprisingly, we find that just test-time batchnorm gives huge performance improvements in BNNs of $21.9\%$ for CIFAR-10-C for the highest corruption level, which is more than twice the performance improvement for ensembles of $10.6\%$. We then go on to develop ShiftMatch which gives even larger improvements of $24.6\%$ for BNNs and $12.4\%$ for ensembles. While ShiftMatch bears some superficial similarities to test-time batchnorm, it is in reality quite different. In particular, on convolutional neural networks, batchnorm matches only the means and variances for each channel. In contrast, ShiftMatch matches the spatial covariances statistics for each channel separately, allowing us to correct complex, spatially extended corruptions, and to perfectly correct certain simpler corruptions such as Gaussian blur (Sec. 4.6). We find ShiftMatch gives considerable performance improvements over test-time batchnorm (Figs. 4,9).

There is work suggesting whitening or transforming the feature correlations (Huang et al., 2018), in part to improve corruption robustness (Roy et al., 2019; Choi et al., 2021). These approaches have at least three key differences from ShiftMatch. First, these approaches whiten at train-time, which dramatically slows down training, and prevents the use of pre-trained networks, regardless of being Bayesian or not. In contrast, ShiftMatch training is exactly equivalent to faster standard (B)NN training, allowing the use of pre-trained models. Second, these papers apply to images with

$C$ channels, width $W$ and height $H$, but whiten only over channels (i.e. using $C \times C$ matrices). However, this ignores the complex *spatial* structure present in most corruptions: we find that by transforming *spatial* correlations we can perfectly remove simple corruptions such as Gaussian blur (Sec. 4.6). As such, we instead transform the spatial correlations of each channel separately (see Fig. 7 for variants). Importantly, these two choices interact: it probably is not possible to work with spatial correlations at training time, because at training time, we only have a few samples in the minibatch (e.g. $N_{\text{mini}} = 128$) to estimate high-dimensional spatial covariances (e.g. for MNIST, these covariance matrices would be $HW \times HW = 784 \times 784$). ShiftMatch largely avoids this issue because it computes the spatial correlations once at the end, using a large amount of training and testing data, giving extremely accurate covariance estimates (see Sec. 4.4 for further details).

There is also a line of work (e.g. Sun et al., 2020; Wang et al., 2020; Zhang et al., 2022) aiming at improving the robustness of pre-trained model by performing self-supervised training at test time. This approach is likely to be too computationally expensive in our setting, as we need to evaluate on many different HMC samples/ensemble members.

Trinh et al. (2022) propose a method for inducing corruption robustness based on a Bayesian interpretation of dropout (Gal & Ghahramani, 2016). This method is very different, in that it proposes a modified training objective, and therefore cannot use gold standard HMC samples.

# 4 METHODS

## 4.1 REINTERPRETING EMPCOV AS A TRAINING-DATA-DEPENDENT LIKELIHOOD

The EmpCov prior over weights (Eq. 3) induces a distribution over $\mathbf{f} = \mathbf{X}\mathbf{w} \in \mathbb{R}^P$, the value of a single output feature for all $P$ inputs (see Appendix A for a full derivation),

$$P_{\text{EmpCov}}(\mathbf{f}|\mathbf{X}_{\text{test}}) = \mathcal{N}\left(\mathbf{f}; \mathbf{0}, \tfrac{1}{MN_{\text{train}}}\mathbf{X}_{\text{test}}\left(\mathbf{X}_{\text{train}}^\top \mathbf{X}_{\text{train}}\right)\mathbf{X}_{\text{test}}^\top\right) \tag{5}$$

We can obtain an equivalent distribution over $\mathbf{f}$ by first transforming the inputs using ShiftEmpCov,

$$\text{ShiftEmpCov}(\mathbf{X}_{\text{test}}; \mathbf{X}_{\text{train}}) = \mathbf{X}_{\text{test}}\left(\tfrac{1}{N_{\text{train}}}\mathbf{X}_{\text{train}}^\top \mathbf{X}_{\text{train}}\right)^{1/2}. \tag{6}$$

In the arguments to ShiftEmpCov, the "input", $\mathbf{X}_{\text{test}}$ is to the left of the semicolon, and the quantities used purely for normalization, $\mathbf{X}_{\text{train}}$, are to the right of the semicolon. Computing $\mathbf{f}$ from these transformed inputs, $\mathbf{f} = \text{ShiftEmpCov}(\mathbf{X}_{\text{test}}; \mathbf{X}_{\text{train}})\mathbf{w}$, and using a standard, isotropic prior, $P(\mathbf{w}) = \mathcal{N}\left(0, \tfrac{1}{M}\mathbf{I}\right)$, recovers the EmpCov distribution over $\mathbf{f}$ (Eq. 5). (For details of the full equivalence using the function-space viewpoint, see Appendix B). Importantly, ShiftEmpCov moves the explicit training-data-dependence from the prior to the likelihood,

$$P_{\text{ShiftEmpCov}}(\mathbf{y}_{\text{train}}|\mathbf{X}_{\text{train}}, \mathbf{w}) = \text{Categorical}(\mathbf{y}_{\text{train}}; \pi_{\text{ShiftEmpCov}; \mathbf{X}_{\text{train}}}(\mathbf{X}_{\text{train}}, \mathbf{w})), \tag{7a}$$

$$P_{\text{ShiftEmpCov}}(\mathbf{y}_{\text{test}}|\mathbf{X}_{\text{test}}, \mathbf{w}, \mathbf{X}_{\text{train}}) = \text{Categorical}(\mathbf{y}_{\text{test}}; \pi_{\text{ShiftEmpCov}; \mathbf{X}_{\text{train}}}(\mathbf{X}_{\text{test}}, \mathbf{w})). \tag{7b}$$

Here, $\pi_{\text{ShiftEmpCov}; \mathbf{X}_{\text{train}}}$ is a function parameterised by a neural network with ShiftEmpCov at the input layer. Critically, this function (in particular, the normalization) depends on $\mathbf{X}_{\text{train}}$, giving a training-data dependent likelihood. We can then apply the function to arbitrary input data, (which could be training, $\mathbf{X}_{\text{train}}$, test, $\mathbf{X}_{\text{test}}$, or something else), and weights, $\mathbf{w}$, giving $\pi_{\text{ShiftEmpCov}; \mathbf{X}_{\text{train}}}(\cdot, \mathbf{w})$. Importantly, we are using exactly the same function, $\pi_{\text{ShiftEmpCov}; \mathbf{X}_{\text{train}}}(\cdot, \mathbf{w})$ for the train and test likelihoods: the only unusual thing is that the function itself (i.e. the likelihood) depends on the training data, which is analogous to how the prior depended on the training data in the original EmpCov setup.

## 4.2 DEVELOPING IMPROVED TRAINING-DATA-DEPENDENT LIKELIHOODS

Motivated by EmpCov's equivalent training-data-dependent likelihoods and priors, we considered other training-data-dependent likelihoods that might resolve issues with EmpCov priors. In particular, we designed ShiftMatch as a training-data-dependent likelihood that is capable of reusing samples from pre-trained networks. The ShiftMatch likelihood is built upon the ShiftMatch transformation

$$\text{ShiftMatch}(\mathbf{H}_{\text{test}}; \mathbf{H}_{\text{train}}) = \tilde{\mathbf{H}}_{\text{test}}\left(\tfrac{1}{N_{\text{test}}}\tilde{\mathbf{H}}_{\text{test}}^\top \tilde{\mathbf{H}}_{\text{test}}\right)^{-1/2}\left(\tfrac{1}{N_{\text{train}}}\tilde{\mathbf{H}}_{\text{train}}^\top \tilde{\mathbf{H}}_{\text{train}}\right)^{1/2} + \mathbf{M}_{\text{train}}, \tag{8}$$

where $\tilde{\mathbf{H}}_{\text{test}}$ and $\tilde{\mathbf{H}}_{\text{train}}$ are mean-subtracted versions of the test and training inputs/features, $\mathbf{M}_{\text{train}}$ is the training mean, and we use the symmetric matrix square root, (Huang et al., 2018). ShiftMatch matches the mean and covariance of the test data to that of the training data; for the covariances,

$$\frac{1}{N_{\text{test}}} \left(\text{ShiftMatch}(\mathbf{H}_{\text{test}}; \mathbf{H}_{\text{train}}) - \mathbf{M}_{\text{train}}\right) \left(\text{ShiftMatch}(\mathbf{H}_{\text{test}}; \mathbf{H}_{\text{train}}) - \mathbf{M}_{\text{train}}\right)^{\top} = \frac{1}{N_{\text{train}}} \tilde{\mathbf{H}}_{\text{train}}^{\top} \tilde{\mathbf{H}}_{\text{train}} \tag{9}$$

As such, applying ShiftMatch to the training data leaves it unchanged,

$$\text{ShiftMatch}(\mathbf{H}_{\text{train}}; \mathbf{H}_{\text{train}}) = \tilde{\mathbf{H}}_{\text{train}} + \mathbf{M}_{\text{train}} = \mathbf{H}_{\text{train}}, \tag{10}$$

allowing us to train the network without including the ShiftMatch layers. As the priors over weights are also the same isotropic Gaussian priors used in standard BNNs, we are able to use e.g. pre-trained gold-standard samples from HMC-trained BNNs (Izmailov et al., 2021b). A summary of the full ShiftMatch algorithm can be found in Appendix E Alg.1. Overall, ShiftMatch likelihoods have the same basic form as the ShiftEmpCov likelihoods (Eq. 7),

$$\text{P}_{\text{ShiftMatch}} \left(\mathbf{y}_{\text{train}} | \mathbf{X}_{\text{train}}, \mathbf{w}\right) = \text{Categorical}(\mathbf{y}_{\text{train}}; \pi_{\text{ShiftMatch}; \mathbf{X}_{\text{train}}}(\mathbf{X}_{\text{train}}, \mathbf{w})) \tag{11a}$$

$$\text{P}_{\text{ShiftMatch}} \left(\mathbf{y}_{\text{test}} | \mathbf{X}_{\text{test}}, \mathbf{w}, \mathbf{X}_{\text{train}}\right) = \text{Categorical}(\mathbf{y}_{\text{test}}; \pi_{\text{ShiftMatch}; \mathbf{X}_{\text{train}}}(\mathbf{X}_{\text{test}}, \mathbf{w})) \tag{11b}$$

The key difference is that for the training data, the likelihoods are equal to those in standard BNNs,

$$\pi_{\text{ShiftMatch}; \mathbf{X}_{\text{train}}}(\mathbf{X}_{\text{train}}, \mathbf{w}) = \pi(\mathbf{X}_{\text{train}}, \mathbf{w}) \tag{12a}$$

$$\text{P}_{\text{ShiftMatch}} \left(\mathbf{y}_{\text{train}} | \mathbf{X}_{\text{train}}, \mathbf{w}\right) = \text{P} \left(\mathbf{y}_{\text{train}} | \mathbf{X}_{\text{train}}, \mathbf{w}\right) \tag{12b}$$

where standard BNN likelihoods and NNs are denoted by $\text{P}$ and $\pi$ without any subscript. This implies that ShiftMatch posteriors are exactly equal to posteriors in a BNN with standard likelihoods, allowing us to use samples of the weights from pre-trained networks,

$$\text{P}_{\text{ShiftMatch}} \left(\mathbf{w} | \mathbf{y}_{\text{train}}, \mathbf{X}_{\text{train}}\right) \propto \text{P}_{\text{ShiftMatch}} \left(\mathbf{y}_{\text{train}} | \mathbf{X}_{\text{train}}, \mathbf{w}\right) \text{P} \left(\mathbf{w}\right)$$
$$= \text{P} \left(\mathbf{y}_{\text{train}} | \mathbf{X}_{\text{train}}, \mathbf{w}\right) \text{P} \left(\mathbf{w}\right) \propto \text{P} \left(\mathbf{w} | \mathbf{y}_{\text{train}}, \mathbf{X}_{\text{train}}\right). \tag{13}$$

### 4.3 STRUCTURED COVARIANCE ESTIMATION IN CNNS

The ShiftMatch operation shown in Eq. (8) is appropriate for fully-connected networks with a relatively small number of features. However, this approach cannot be applied directly to CNNs, because the feature maps are of size $CHW$, where $C$ is the number of channels, $H$ is the height and $W$ is the width. Thus, we end up with huge $CHW \times CHW$ covariance matrices, which presents computational difficulties (e.g. the matrix square roots require $\mathcal{O}(C^3 H^3 W^3)$ compute).

Instead, we need to structure our covariance estimates to improve accuracy and speed up matrix computations. We begin by choosing to match the spatial covariance for each channel separately, which would require us to estimate $C$ separate $HW \times HW$ covariance matrices. Even then, this poses difficulties because images or feature maps can have very large numbers of pixels. Thus, we need to impose further spatial structure, and we choose to treat the covariance along the height and width separately, using a Kronecker factored covariance (Martens & Grosse, 2015), so we end up with a $H \times H$ matrix and a $W \times W$ matrix for each channel (see Appendix Fig. 7 for other variants).

### 4.4 COMPARISON WITH BATCHNORM

ShiftMatch does bear some minor resemblance to batchnorm, in that it matches the feature moments to prespecified values. Indeed, inspired by ShiftMatch, we could develop a "spatial batchnorm", a generalisation of batchnorm that matches spatial covariances of feature maps to learned values. However, even spatial batchnorm would ultimately be quite different from ShiftMatch. In particular, batchnorm is usually applied only at train-time, and turned off at test-time (though it is sometimes left on at test-time, where it can improve robustness to corruption Nado et al., 2020; Schneider et al., 2020). However, ShiftMatch is the opposite in that it is applied only at test-time; at train-time, ShiftMatch is identical to a standard NN without extra normalization layers (Eq. 10). Thus, ShiftMatch allows us to decouple test time from train time normalization, which turns out to be a powerful idea. In particular, ShiftMatch has two key advantages over a spatial batchnorm. First, the required spatial covariance matrices are complex, high-dimensional objects, which require much more than a single minibatch to estimate accurately. With ShiftMatch, it is straightforward to aggregate over minibatches,

as we estimate these matrices after training the weights using any available training and test data. Moreover, as we compute the covariances after training, we can use large batchsizes (e.g. we can fit a batch of 1000 for ImageNet even on a single 2080ti with 11GB of memory), because we do not need to retain feature maps at all layers for use in the backward pass (for further discussion, see e.g. Sander et al., 2021). In contrast, training batches used in a spatial batchnorm would need to be much smaller (often 16 examples for ImageNet on smaller GPUs), which can give very inaccurate or even low-rank, non-invertible estimates of the covariance matrices. Second, ShiftMatch trains much faster than a spatial batchnorm. In particular, even after carefully simplifying the covariance matrices, computing the matrix square roots at every layer at every gradient step is still very slow. In contrast, in ShiftMatch, we only do the matrix square roots once after training. For instance, it took us around 0.35s for a forward pass without spatial batchnorm, which contrasts with 0.77s for a forward pass with spatial batchnorm using a mini-batch of 128 inputs for CIFAR-10.

## 4.5 TRAINING DATA AT TEST TIME?

One potential disadvantage of ShiftMatch is the requirement for training data to compute moments, which is sometimes not provided for pre-trained models. However, this requirement is mitigated by three factors. First, if ShiftMatch becomes commonplace, then the necessary training statistics could be released along with model weights. This is in principle exactly the same as capturing the training means and variances in batchnorm layers. Second, you do not need to use the whole training set. You only need enough datapoints to get a reasonable estimate of the covariance matrices (see Fig. 5, where performance saturates after using only 10,000 training images, less than 1% of the full dataset). Third, we may not even need training inputs, as long as we can get proxy inputs that are close enough. Fig. 1, shows only a small performance decrease when we do not use the training dataset to obtain the training statistics, but instead we use the entirely separate CIFAR10.1 captured by Recht et al. (2019).

## 4.6 SHIFTMATCH PERFECTLY REMOVES STATIONARY LINEAR CORRUPTIONS

ShiftMatch retains all the useful theoretical properties of EmpCov priors, in suppressing pathologies that arise in BNNs when there is low-input variance along a direction in the training but not in the corrupted test data (see Sec. 1 and Izmailov et al., 2021a for more details). ShiftMatch also has considerable theoretical advantages over EmpCov priors, in that certain simpler corruptions can be removed perfectly (Fig. 2). In particular, Shift-Match can perfectly remove linear spatially stationary corruptions such as Gaussian blur (in the limit of infinitely many samples). Without loss of generality, consider $\mathbf{X}_{\text{train}} \in \mathbb{R}^{N_{\text{train}} \times HW}$ as zero-mean training images with a single input channel (e.g. a grayscale image). We have uncorrupted test data, $\mathbf{X}_{\text{uncor}} \in \mathbb{R}^{N_{\text{test}} \times HW}$ which is corrupted with a stationary linear spatial operation, represented by $\mathbf{S} \in \mathbb{R}^{HW \times HW}$,

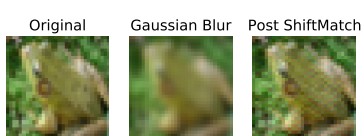

Figure 2: An example of ShiftMatch applied on Gaussian blur corrupted image.

$$\mathbf{X}_{\text{cor}} = \mathbf{X}_{\text{uncor}}\mathbf{S} \in \mathbb{R}^{N_{\text{test}} \times HW}. \tag{14}$$

Taking the SVD of $\mathbf{X}_{\text{train}}$, $\mathbf{X}_{\text{uncor}}$ and $\mathbf{S}$, we get,

$$\mathbf{X}_{\text{train}} = \mathbf{U}_{\text{train}}\mathbf{D}\mathbf{V}^\top \qquad \mathbf{X}_{\text{uncor}} = \mathbf{U}_{\text{test}}\mathbf{D}\mathbf{V}^\top \qquad \mathbf{S} = \mathbf{V}\mathbf{D}_{\text{s}}\mathbf{V}^\top. \tag{15}$$

Here, we have used the fact that the right singular vectors of stationary signals such as images is a Fourier basis, $\mathbf{V}$, and $\mathbf{D}$ is the power-spectrum of uncorrupted images (Olshausen & Field, 1996; Hernandez et al., 2018), which is the same for the uncorrupted training data, $\mathbf{X}_{\text{train}}$ and uncorrupted test data, $\mathbf{X}_{\text{uncor}}$. Further, the eigenbasis of stationary transformations such as Gaussian blur is again the Fourier basis, $\mathbf{V}$. Thus, the SVD of the corrupted test data is,

$$\mathbf{X}_{\text{cor}} = \mathbf{X}_{\text{uncor}}\mathbf{S} = \mathbf{U}_{\text{test}}\mathbf{D}\mathbf{V}^\top\mathbf{V}\mathbf{D}_{\text{s}}\mathbf{V}^\top = \mathbf{U}_{\text{test}}\mathbf{D}\mathbf{D}_{\text{s}}\mathbf{V}^\top. \tag{16}$$

Critically, applying ShiftMatch to the corrupted test data exactly recovers the uncorrupted test data,

$$\text{ShiftMatch}\left(\mathbf{X}_{\text{cor}}; \mathbf{X}_{\text{train}}\right) = \mathbf{X}_{\text{cor}} \left(\tfrac{1}{N_{\text{test}}}\mathbf{X}_{\text{cor}}\mathbf{X}_{\text{cor}}\right)^{-1/2} \left(\tfrac{1}{N_{\text{train}}}\mathbf{X}_{\text{train}}\mathbf{X}_{\text{train}}\right)^{1/2} \tag{17}$$

$$= \mathbf{U}_{\text{test}}\left(\mathbf{D}\mathbf{D}_{\text{s}}\right)\mathbf{V}^\top \left(\mathbf{V}\left(\mathbf{D}\mathbf{D}_{\text{s}}\right)^{-1}\mathbf{V}\right)\left(\mathbf{V}\mathbf{D}\mathbf{V}^\top\right) = \mathbf{U}_{\text{test}}\mathbf{D}\mathbf{V}^\top = \mathbf{X}_{\text{uncor}}$$

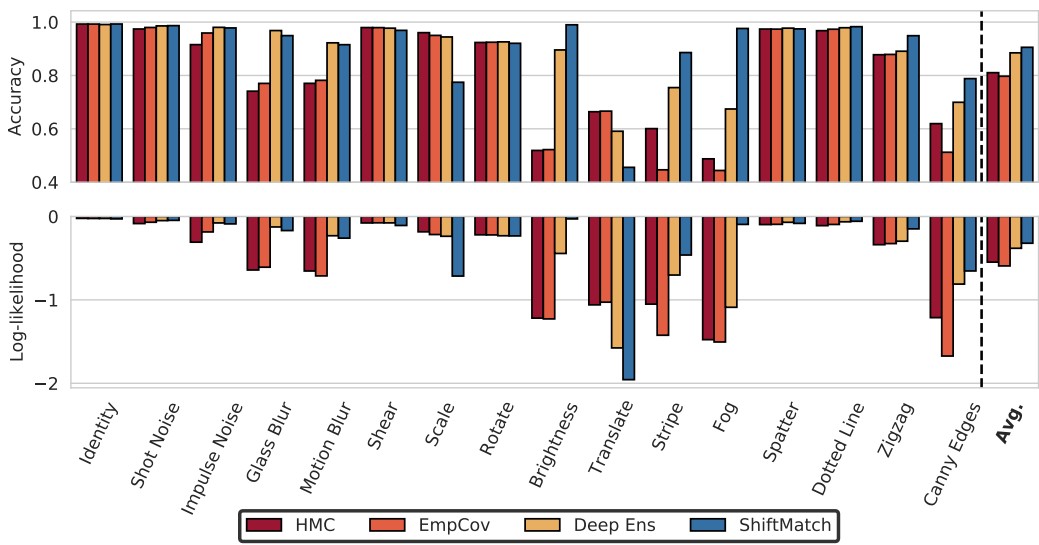

Figure 3: Performance of HMC with isotropic priors (HMC) and EmpCov priors (EmpCov), deep ensembles and ShiftMatch on MNIST-C under 15 different corruption types. The left-most column gives the performance on clean MNIST while the right-most column gives the performance averaged across the 15 corruptions. ShiftMatch performs better overall, but uses additional information from a minibatch of corrupted test datapoints, which is not used by the other methods.

## 5  EXPERIMENTS

First, we applied ShiftMatch to the HMC samples from Izmailov et al. (2021b) for a large-scale Bayesian ResNet trained on CIFAR-10 and tested on CIFAR-10-C (Hendrycks & Dietterich, 2019), and compare to the baselines in Izmailov et al. (2021b). ShiftMatch gives a considerable improvement over using the raw HMC samples and over all baselines (including e.g. ensembles). Second, we show that ShiftMatch performs better than EmpCov priors on small CNNs from Izmailov et al. (2021a) trained on MNIST and tested on MNIST-C(Mu & Gilmer, 2019). Third, we find that ShiftMatch improves over test-time batchnorm when we use HMC samples from Izmailov et al. (2021b), and when we consider non-Bayesian baselines (SGD and ensembles; Fig. 4). Fourth, we show that ShiftMatch can be applied on a large pre-trained non-Bayesian network, where it improved performance on ImageNet relative to test-time batchnorm. Finally, we evaluate the performance of ShiftMatch on an OOD detection task and we find it has no obvious beneficial or harmful effects.

**CIFAR-10-C baselines from Izmailov et al. (2021b).** We began by applying ShiftMatch to the gold-standard HMC samples from a BNN trained on CIFAR-10 from Izmailov et al. (2021b), and comparing to the baselines from that paper (Fig. 1, Appendix C, Fig. 6). They used a ResNet-20 with only 40,960 of the 50,000 training samples (in order to "evenly share the data across the TPU devices"), and to ensure deterministic likelihood evaluations (which is necessary for HMC), turned off data augmentation and data subsampling (i.e. full batch training), and used filter response normalization (FRN) (Singh & Krishnan, 2020) rather than batch normalization (Ioffe & Szegedy, 2015). Despite these stringent constraints, their network achieved excellent performance, with greater than 90% accuracy on the clean test set. For further details of how these baselines were generated, please see Izmailov et al. (2021b). We found that ShiftMatch considerably improved over all baselines included in that paper, including the raw HMC samples themselves, and deep ensembles.

**MNIST-C baselines from Izmailov et al. (2021a).** To compare against EmpCov priors, we were forced to use smaller-scale networks and datasets, because that is all that is provided in the EmpCov paper (Izmailov et al., 2021a). Presumably, this was because they wanted gold-standard HMC samples, but no longer had access to the large scale compute in Izmailov et al. (2021b). In particular, they used smaller models such as multilayer perceptron and a LeNet-5 (LeCun et al., 1998), which contains 2 convolutional layers followed by 3 fully-connected layers. We used the HMC samples for

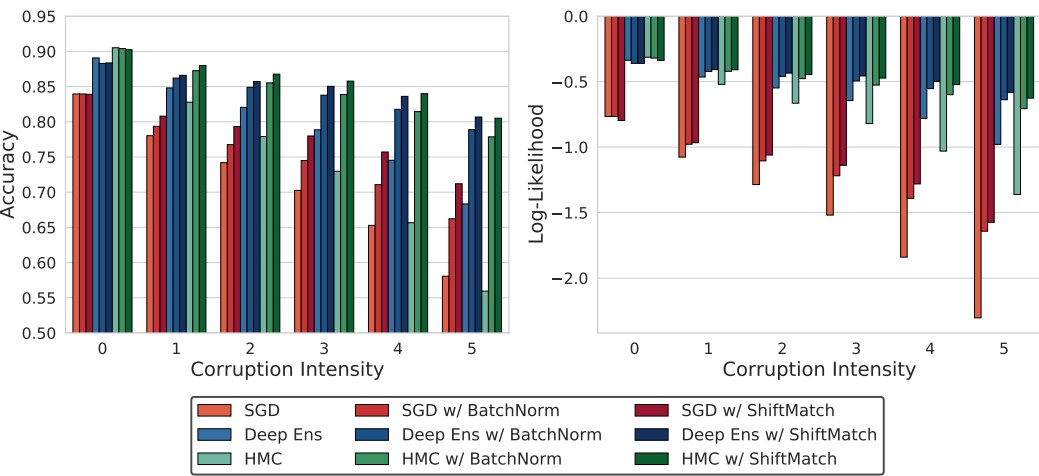

Figure 4: All combinations of SGD (red, left three bars), ensembles (blue, middle three bars) and HMC (green, right three bars) with no normalization, batchnorm and ShiftMatch for CIFAR-10-C. Note that ShiftMatch and test-time batchnorm use information from a minibatch of corrupted inputs, while the plain methods do not.

LeNet trained on MNIST under standard Gaussian prior and the EmpCov prior provided by Izmailov et al., 2021a to evaluate different methods on MNIST-C (Mu & Gilmer, 2019). Additionally, we used their code to get an ensemble of 50 independently initialised models trained using SGD with momentum as the combined with cosine-annealed learning rate decay and a batchsize of 80. Again, we found that ShiftMatch performed better than the other methods, including HMC with standard Gaussian and EmpCov priors, and non-Bayesian baselines such as ensembles (Fig. 3). Note the poor performance of ShiftMatch on translate, which can be understood at least in part because translations are not stationary transformations that can be exactly reversed by a ShiftMatch (Sec. 4.6). That said, robustness to translations is usually induced by data augmentation which was not used when Izmailov et al. (2021a) were training their networks.

**Test-time batchnorm with BNNs, and ShiftMatch with non-Bayesian methods.** Above, we focused on combining ShiftMatch with BNNs, and we found that these models performed better than previous approaches used in the BNN literature (Izmailov et al., 2021b;a). However, ShiftMatch could also be applied to non-Bayesian networks, and test-time batchnorm could be applied to BNNs. We therefore considered all possible combinations of ShiftMatch and batchnorm with SGD, ensembles and HMC (Fig. 4). To ensure a fair comparison, we used networks, HMC samples, training code and experimental settings from Izmailov et al. (2021b). To be more specific, for SGD, we used a network architecture identical to that of HMC: A ResNet-20 with batchnorm replaced by FRN. The ensemble model was constructed using 50 SGD models with different initialisations. As the underlying network did not have batchnorm layers, we used a variant of ShiftMatch to match the mean and the variance of each channel with the training sample statistics (see Fig. 9 for full batchnorm).

In all cases we found that test-time batchnorm improved over plain models, and ShiftMatch improved over test-time batchnorm. Confirming results from Izmailov et al. (2021b), we found that HMC without either test-time batchnorm or ShiftMatch performed very poorly: worse than ensembles for any level corruption. In contrast, with ShiftMatch, HMC and deep ensembles performed very similarly. Indeed, ShiftMatch gave spectacular improvements in accuracy of 24.6% for HMC, and smaller improvements of 12.4% for ensembles (Appendix Tab. 2).

**Scaling to ImageNet with non-Bayesian ShiftMatch** ShiftMatch can be applied to Bayesian or non-Bayesian pre-trained networks, and can readily be applied to checkpoints from larger pre-trained models. To demonstrate this, we took an ImageNet checkpoint from Keras Applications[2] and applied ShiftMatch. Note that this network included batchnorm in training; to combine ShiftMatch with

---

[2]https://keras.io/api/applications/

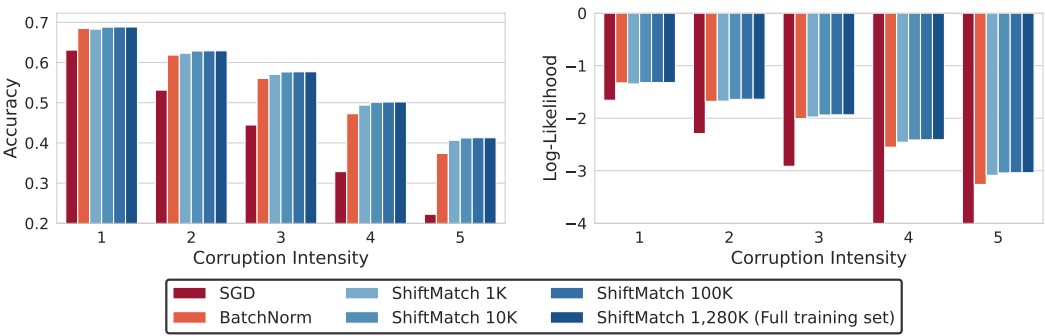

Figure 5: Performance of plain SGD, test-time BatchNorm and ShiftMatch on ImageNet-C (Hendrycks & Dietterich, 2019). The number after ShiftMatch indicates the number of training points (out of the 1, 280K training samples) used for estimating the training statistics. All results presented use a pre-trained ResNet-50 as the backbone.

Table 1: OOD detection performance of models trained on CIFAR-10, taking CIFAR-100 and SVHN as OOD datasets. Note the arrows indicate the "better" direction (e.g. so lower FPR95 is better).

| | AUROC ↑ | | | | FPR95 ↓ | | | |
|---|---|---|---|---|---|---|---|---|
| OOD dataset | SGD | Deep Ens | HMC | ShiftMatch | SGD | Deep Ens | HMC | ShiftMatch |
| CIFAR-100 | 0.780 | 0.830 | **0.851** | 0.848 | 0.579 | 0.490 | **0.457** | 0.480 |
| SVHN | 0.713 | 0.788 | 0.874 | **0.920** | 0.625 | 0.453 | 0.299 | **0.281** |

batchnorm, we added ShiftMatch after the batchnorm layer (see Appendix F for further details). We found that ShiftMatch again offers considerable gains over just using batchnorm, especially at the higher corruption intensities (Fig. 5). The test statistics are computed using a batch of 1,000 images. The training statistics are computed using a random subset of 1,000, 10,000 or 100,000 training images, or the full dataset. We saw a slight reduction in performance when using only 1,000 training images, but no differences otherwise.

**Effect of ShiftMatch on uncertainty quantification** While BNNs have poor performance under corruption, they still provide high-quality uncertainty estimates that can benefit down-stream tasks, such as detecting OOD inputs (Wang & Aitchison, 2021; Kristiadi et al., 2022). We might think that ShiftMatch would be detrimental to OOD detection accuracy, as it would make OOD samples more similar to training samples. We consider models trained on CIFAR-10, and asked whether we could identify inputs from CIFAR-100 or SVHN. We used SGD, ensembles and HMC with and without ShiftMatch. For all the models, we used the Izmailov et al. (2021b) setting as in Figs. 1–4. We used the maximum softmax probability (Hendrycks & Gimpel, 2017) of the (ensembled) predictive distribution as the score and we used AUROC and FPR95 as the performance metric. We found that HMC and ShiftMatch always perform better than ensembles and SGD (Tab. 1). The performance of HMC with and without ShiftMatch was very similar; HMC *without* ShiftMatch performed slightly better on CIFAR-100, and HMC *with* ShiftMatch performed slightly better on SVHN. These experiments show that any detrimental effect of ShiftMatch on OOD detection performance is likely to be small.

## 6 CONCLUSION

We developed a new training-data-dependent likelihood for robustness to corruption for BNNs, called ShiftMatch. ShiftMatch has the key limitation that, like keeping batchnorm on at test-time but unlike EmpCov priors, it uses information from a minibatch of similarly corrupted test points. We found that ShiftMatch indeed offered considerable improvements over past approaches to robustness to corruption in BNNs, both in CIFAR-10-C (Fig. 1) and in MNIST-C (Fig. 3), and even can give improvements over test-time batchnorm when used in non-Bayesian settings (Figs. 4,9).

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

## A    DERIVATION OF THE EMPCOV DISTRIBUTION OVER FEATURES

We have $\mathbf{f} = \mathbf{X}_{\text{test}}\mathbf{w}$, with the EmpCov prior over weights, Eq. (3). Our goal is to compute, the implied distribution, $\mathrm{P}\left(\mathbf{f}|\mathbf{X}_{\text{test}}\right)$. As $\mathbf{f} = \mathbf{X}_{\text{test}}\mathbf{w}$ is a linear combination of Gaussian $\mathbf{w}$ with fixed $\mathbf{X}_{\text{test}}$, we know $\mathrm{P}\left(\mathbf{f}|\mathbf{X}_{\text{test}}\right)$ that is multivariate Gaussian,

$$\mathrm{P}\left(\mathbf{f}|\mathbf{X}_{\text{test}}\right) = \mathcal{N}\left(\mathbf{f}; \boldsymbol{\mu}, \boldsymbol{\Sigma}\right). \tag{18}$$

Therefore to characterise the full distribution, $\mathrm{P}\left(\mathbf{f}|\mathbf{X}\right)$, we just need to compute the expectation, $\boldsymbol{\mu}$, and covariance, $\boldsymbol{\Sigma}$. The mean is zero,

$$\boldsymbol{\mu} = \mathrm{E}\left[\mathbf{f}|\mathbf{X}_{\text{test}}\right] = \mathbf{X}_{\text{test}}\,\mathrm{E}\left[\mathbf{w}\right] = \mathbf{0} \tag{19}$$

as $\mathrm{E}\left[\mathbf{w}\right] = \mathbf{0}$ (Eq. 3). And the covariance is,

$$\boldsymbol{\Sigma} = \mathrm{E}\left[\mathbf{f}\mathbf{f}^\top|\mathbf{X}\right] = \mathbf{X}_{\text{test}}\,\mathrm{E}\left[\mathbf{w}\mathbf{w}^\top\right]\mathbf{X}_{\text{test}}^\top = \tfrac{1}{CN_{\text{train}}}\mathbf{X}_{\text{test}}\left(\mathbf{X}_{\text{train}}^\top\mathbf{X}_{\text{train}}\right)\mathbf{X}_{\text{test}}^\top \tag{20}$$

as $\mathrm{E}\left[\mathbf{w}\mathbf{w}^T\right] = \tfrac{1}{CN_{\text{train}}}\mathbf{X}_{\text{train}}^\top\mathbf{X}_{\text{train}}$. Substituting $\boldsymbol{\mu}$ (Eq. 19) and $\mathbf{S}$ (Eq. 20) into Eq. (18) gives Eq. (5).

## B    EQUIVALENCE OF POSTERIORS UNDER EMPCOV PRIORS AND SHIFTEMPCOV LIKELIHOODS

The key claim in Sec. 4.1 is that models with EmpCov priors and ShiftEmpCov likelihoods have equivalent behaviour, because the distribution of the outputs of a layer with an EmpCov prior and a ShiftEmpCov likelihood are the same.

To formalise this notion of equivalence, we need to look at a function-space viewpoint (e.g. Rudner et al., 2021). This viewpoint works with the distribution over the logits (or neural network outputs) $\mathbf{L}$. The logits, $\mathbf{L}$, depend on the inputs, $\mathbf{X}$ and the neural network weights, $\mathbf{w}$. Of course, $\mathbf{L}$ is usually a deterministic function of $\mathbf{X}$ and $\mathbf{w}$. The function space viewpoint (e.g. [1]) induces a distribution over functions (i.e. logits or network outputs, $L$) by marginalising the prior over weights,

$$\mathrm{P}\left(\mathbf{L}|\mathbf{X}\right) = \int d\mathbf{w}\,\mathrm{P}\left(\mathbf{L}|\mathbf{w}, \mathbf{X}\right)\mathrm{P}\left(\mathbf{X}\right). \tag{21}$$

While explicitly computing $\mathrm{P}\left(\mathbf{L}|\mathbf{X}\right)$ is usually intractable, this viewpoint is very useful for establishing theoretical properties. In particular, as the class labels depend only on the logits/network outputs, we can write the full function-space graphical model as,

$$\mathbf{X} \rightarrow \mathbf{L} \rightarrow \mathbf{y}. \tag{22}$$

Here, we combine the training and test points, $\mathbf{X} = \left(\mathbf{X}_{\text{train}}, \mathbf{X}_{\text{test}}\right)$, $\mathbf{L} = \left(\mathbf{L}_{\text{train}}, \mathbf{L}_{\text{test}}\right)$ and $\mathbf{y} = \left(\mathbf{y}_{\text{train}}, \mathbf{y}_{\text{test}}\right)$. The prior over logits, $\mathrm{P}\left(\mathbf{L}|\mathbf{X}\right)$ is the same in both EmpCov and ShiftEmpCov viewpoints, because the prior distribution of the outputs of each layer is the same. Likewise, the softmax likelihood $\mathrm{P}\left(\mathbf{y}|\mathbf{L}\right)$ is again the same in both EmpCov and ShiftEmpCov viewpoints. As the priors are exactly equivalent under the two viewpoints, the posteriors over $\mathbf{L}$ and $\mathbf{y}_{\text{test}}$ must also be equivalent, and we can go further and explicitly write out the equivalent posteriors. The posterior over train and test logits, $\mathbf{L}$, conditioned on all inputs, $\mathbf{X}$ and training labels, $\mathbf{y}_{\text{train}}$, is

$$\mathrm{P}\left(\mathbf{L}|\mathbf{X}, \mathbf{y}_{\text{train}}\right) \propto \mathrm{P}\left(\mathbf{y}_{\text{train}}|\mathbf{L}_{\text{train}}\right)\mathrm{P}\left(\mathbf{L}|\mathbf{X}\right). \tag{23}$$

Remember that $\mathbf{L} = \left(\mathbf{L}_{\text{train}}, \mathbf{L}_{\text{test}}\right)$ and $\mathbf{X} = \left(\mathbf{X}_{\text{train}}, \mathbf{X}_{\text{test}}\right)$. Again, this posterior is the same for the EmpCov and ShiftEmpCov viewpoints, because the prior, $\mathrm{P}\left(\mathbf{L}|\mathbf{X}\right)$, is the same, and because the softmax likelihood, $\mathrm{P}\left(\mathbf{y}_{\text{train}}|\mathbf{L}_{\text{train}}\right)$ is the same. From the posterior over logits, we can obtain the predictive distribution over test labels,

$$\mathrm{P}\left(\mathbf{y}_{\text{test}}|\mathbf{X}, \mathbf{y}_{\text{train}}\right) = \int d\mathbf{L}\,\mathrm{P}\left(\mathbf{y}_{\text{test}}|\mathbf{L}\right)\mathrm{P}\left(\mathbf{L}|\mathbf{X}, \mathbf{y}_{\text{train}}\right), \tag{24}$$

and this is again equivalent because the posterior over logits is equivalent, and the softmax likelihood is equivalent.

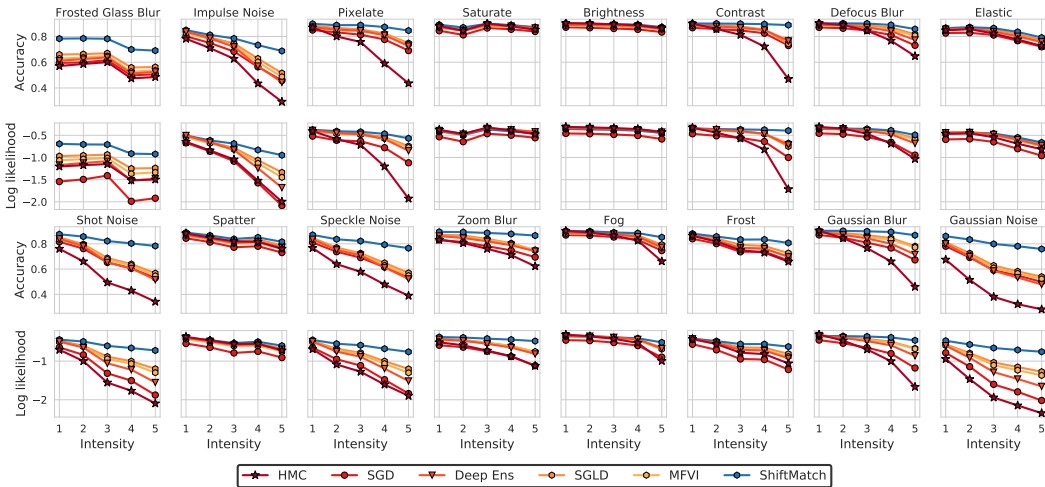

Figure 6: Performance of HMC, SGD, Deep Ensemble, SGLD, Mean-field VI (from dark red to yellow) and ShiftMatch (blue) on CIFAR-10-C under 16 different corruption types with 5 intensity levels.

## C   FURTHER EXPERIMENTAL RESULTS

Here, we present full performance across different corruptions for HMC on CIFAR-10-C and other baselines from (Izmailov et al., 2021b) (Fig. 6). Tab. 2 gives performance differences with and without ShiftMatch, and Tab. 3 gives performance differences with and without test-time batchnorm, for the usual Izmailov et al. (2021b) setting. Note that HMC benefits far more than non-Bayesian methods, with improvements often being more than double those for ensembles.

ShiftMatch did take slightly more time: 1.85s for a minibatch of 10,000 points as opposed to 1.15s without ShiftMatch. This tradeoff is likely acceptable due to the very large improvements in performance.

## D   ABLATION STUDY FOR THE DESIGN OF SHIFTMATCH.

We performed an ablation study to choose the design of ShiftMatch (Fig. 7). EmpCov priors might suggest that applying ShiftMatch on the input alone might be sufficient. However, we found applying ShiftMatch only on the input data had performance much worse than other ShiftMatch variants that matched intermediate features, which emphasised the importance of correcting corrupts at all layers. Second, we considered modifying the spatial covariances. In particular, we considered "w/o Kronecker factored covariance", where we directly estimate the $HW \times HW$ covariance matrices, and "channel-joint", where we compute only a single spatial covariance matrix, aggregating information across all channels (we usually have $C$ separate $HW \times HW$ matrices — one for each channel). Both of these modifications performed worse than our approach. Presumably, "w/o Kronecker factored covariance" compromises the ability to obtain accurate covariance estimates, and "channel-joint" compromises the model's ability to match account for different spatial statistics in different channels. Finally, we considered applying ShiftMatch on the post-activation feature, which gave very slightly worse performance than the usual pre-activation approach. Nonetheless, note that all of these sub-optimal variants outperform plain HMC across all corruption intensities.

## E   ALGORITHMS

We summarised the end-to-end inference procedure of ShiftMatch in Alg. 1.

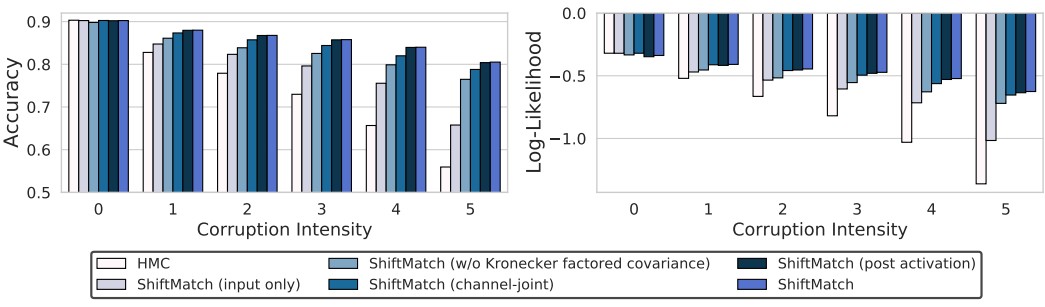

Figure 7: The performance of various variants of ShiftMatch (described in the main text).

Table 2: Accuracy improvement given by using ShiftMatch over the plain method. Note that the benefits of ShiftMatch under HMC are around double the benefits of ShiftMatch for SGD or deep ensembles.

| Corruption Intensity | 1 | 2 | 3 | 4 | 5 |
|---|---|---|---|---|---|
| SGD | 2.8% | 5.1% | 7.7% | 10.4% | 13.1% |
| Deep Ens | 1.8% | 3.7% | 6.2% | 9.1% | 12.4% |
| HMC | 5.2% | 8.9% | 12.8% | 18.3% | 24.6% |

Table 3: Accuracy improvement given by using test-time batchnorm over the plain method.

| Corruption Intensity | 1 | 2 | 3 | 4 | 5 |
|---|---|---|---|---|---|
| SGD | 1.3% | 2.6% | 4.2% | 5.8% | 8.2% |
| Deep Ens | 1.4% | 2.8% | 4.9% | 7.2% | 10.6% |
| HMC | 4.5% | 7.6% | 10.9% | 15.8% | 21.9% |

Table 4: Accuracy improvement given by using ShiftMatch over test-time batchnorm.

| Corruption Intensity | 1 | 2 | 3 | 4 | 5 |
|---|---|---|---|---|---|
| SGD | 1.5% | 2.5% | 3.5% | 4.6% | 4.9% |
| Deep Ens | 0.4% | 0.9% | 1.3% | 1.9% | 1.8% |
| HMC | 0.7% | 1.3% | 1.9% | 2.5% | 2.7% |

## F    INTRODUCING SHIFTMATCH INTO A MODEL WITH BATCHNORM

When we add ShiftMatch to a network which was trained with batchnorm, we add ShiftMatch after the batchnorm layer. In these networks with ShiftMatch, we then have a choice: do we set test-time batchnorm on or off? To answer this question, we tried both (Fig. 8). We found that keeping test-time batchnorm on "SGD w/ SM & BN" performed slightly better than turning off test-time batchnorm "SGD w/ SM".

## G    CIFAR-10 WITH NON-BAYESIAN SHIFTMATCH WITH DATA AUGMENTATION AND TRAINING TIME BATCHNORM.

Remember that Izmailov et al. (2021b) used an unusual setting without data augmentation or train-time batchnorm. We therefore additionally evaluated whether ShiftMatch improved robustness to corruption on CIFAR-10 in the standard setting with data augmentation and train-time batchnorm. In particular, we trained a ResNet-20 using SGD with momentum, with an initial learning rate of $0.05$, a cosine annealing schedule, and a weight decay coefficient of $10^{-4}$. We augmented the training data through random horizontal flip and random crop. To combine ShiftMatch with batchnorm, we

---

**Algorithm 1** End-to-end procedure of ShiftMatch on a pre-trained BNN

---

**Input:** Test set $\mathbf{X}_{\text{test}}$, Training set $\mathbf{X}_{\text{train}}$

  $M$ posterior samples from an $L$-layer BNN $\{\phi_1^m(\cdot), \ldots, \phi_L^m(\cdot)\}, m \in \{1, \ldots, M\}$.

**Step 1. Acquire training statistics**

**for** $m = 1, 2, \cdots, M$ **do**          $\triangleright$ Acquire the training statistics for all $M$ posterior samples.

  $\mathbf{H}_{\text{train}}^0 \leftarrow \mathbf{X}_{\text{train}}$

  **for** $\ell = 0, 1, \cdots, L - 1$ **do**          $\triangleright$ Compute the training statistics $\mathbf{M}_{\text{train}}^{\ell,m}, \mathbf{Q}_{\text{train}}^{\ell,\cdot}$.

    $\mathbf{M}_{\text{train}}^{\ell,m} \leftarrow \text{Average}(\mathbf{H}_{\text{train}}^\ell)$          $\triangleright$ Compute the mean.

    $\tilde{\mathbf{H}}_{\text{train}}^\ell \leftarrow \mathbf{H}_{\text{train}}^\ell - \mathbf{M}_{\text{train}}^\ell$

    $\mathbf{Q}_{\text{train}}^{\ell,m} = \left( \frac{1}{N_{\text{train}}^\ell} (\tilde{\mathbf{H}}_{\text{train}}^\ell)^\top (\tilde{\mathbf{H}}_{\text{train}}^\ell) \right)^{1/2}$          $\triangleright$ Compute the matrix square root.

    $\mathbf{H}_{\text{train}}^{\ell+1} \leftarrow \phi_{\ell+1}(\mathbf{H}_{\text{train}}^\ell)$          $\triangleright$ Forward pass to the next layer.

  **end for**

  Save the training statistics $(\mathbf{M}_{\text{train}}^{\ell,m}, \mathbf{Q}_{\text{train}}^{\ell,m}), \ell \in \{0, \ldots, L-1\}$.

**end for**

**Step 2. Perform Bayesian Model Averaging**

$p \leftarrow \mathbf{0}$          $\triangleright$ Initialise the predicted probabilities

**for** $m = 1, 2, \cdots, M$ **do**          $\triangleright$ Bayesian Model Averaging over $M$ posterior samples

  Load $(\mathbf{M}_{\text{train}}^{\ell,m}, \mathbf{Q}_{\text{train}}^{\ell,m}), \ell \in \{0, \ldots, L-1\}$ from the disk.

  $\mathbf{H}_{\text{test}}^0 \leftarrow \mathbf{X}_{\text{test}}$

  **for** $\ell = 0, 1, \cdots, L - 1$ **do**          $\triangleright$ Match the test-time feature for each layer.

    $\mathbf{M}_{\text{test}}^\ell \leftarrow \text{Average}(\mathbf{H}_{\text{test}}^\ell)$          $\triangleright$ Compute the mean of test feature.

    $\tilde{\mathbf{H}}_{\text{test}}^\ell \leftarrow \mathbf{H}_{\text{test}}^\ell - \mathbf{M}_{\text{test}}^\ell$

    $\mathbf{Q}_{\text{test}}^{*\ell} = \left( \frac{1}{N_{\text{test}}^\ell} \tilde{\mathbf{H}}_{\text{test}}^\top \tilde{\mathbf{H}}_{\text{test}} \right)^{-1/2}$          $\triangleright$ Compute the inverse of symmetric matrix square root.

    $\mathbf{H}_{\text{test}}^{\ell+1} \leftarrow \phi_{\ell+1}^m \left( \tilde{\mathbf{H}}_{\text{test}}^\ell \mathbf{Q}_{\text{test}}^{*\ell} \mathbf{Q}_{\text{train}}^{\ell,m} + \mathbf{M}_{\text{train}}^{\ell,m} \right)$          $\triangleright$ ShiftMatch transformation (Eq. (8)).

  **end for**

  $p \leftarrow p + \frac{1}{M} f(\mathbf{H}_{\text{test}}^L)$, where $f$ can be e.g. softmax.          $\triangleright$ Update the BMA prediction

**end for**

**return** $p$

---

added ShiftMatch after the batchnorm layer (Appendix F). Again, we found that ShiftMatch offered improvements over simply using batchnorm, especially at the higher corruption intensities (Fig. 9).

# H   ANALYSIS OF CALIBRATION ERROR

We have analysed expected calibration error (ECE) in Fig. 10. Broadly, SGD has very poor accuracy and ECE, while ShiftMatch performs best in terms of accuracy, and deep ensembles performs best in terms of ECE. We could understand these results as ShiftMatch trading a bit of ECE for alot of accuracy (see Fig. 10 right). However, comparing ECE is difficult when accuracy differs so dramatically (e.g. for corruption intensity 5; see (Nixon et al., 2019) for further details). As one example, remember that ECE bins the models confidence (e.g. in the range 0.5–0.6), and asks whether the model's probability of being correct is close to the confidence (in this case, the probability of being correct should be about 0.55). When the model is more accurate, more of its probabilities will be close to 0 or close to 1, meaning that there will be fewer samples in the intermediate bins (e.g. see Figure 3 in Garriga-Alonso et al., 2018). Having fewer samples in the intermediate bins implies that there will be larger errors in the calibration curves just due to finite samples, and larger errors in the calibration curve of course means a higher ECE. This is one example where changes in the ECE may occur not due to changes in the model's actual calibration, but due to finite sampling errors.

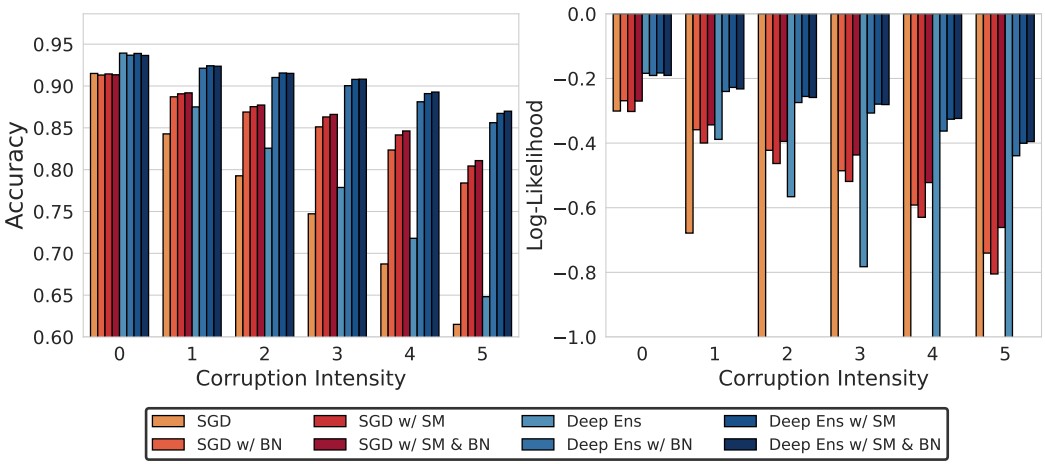

Figure 8: Various combinations of test-time batchnorm and ShiftMatch. "SGD" implies no Shift-Match, and no test-time batchnorm. "SGD w/ BN" implies test-time batchnorm but no ShiftMatch. "SGD w/ SM" implies ShiftMatch but no test-time batchnorm. "SGD w/ SM & BN" implies Shift-Match combined with test-time batchnorm.

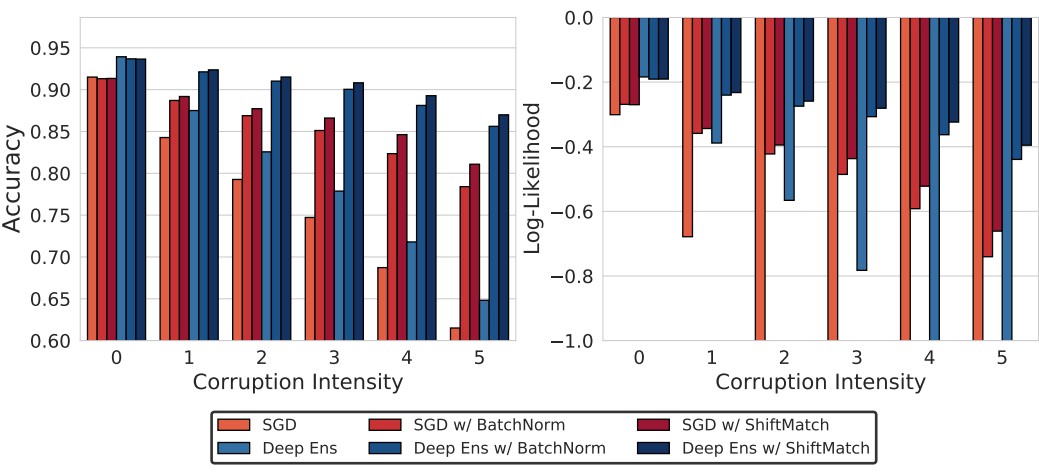

Figure 9: All combinations of SGD (red, left three bars), ensembles (blue, right three bars) with batchnorm and ShiftMatch for CIFAR-10-C with data augmentation and batchnorm at train-time.

## I    PERFORMANCE OF SHIFTMATCH ON MIXTURE OF CLEAN AND CORRUPTED DATA

Here, we considered whether ShiftMatch still improves performance when test minibatches are a 50-50 mix of uncorrupted and fully corrupted (samples of intensity 1-5). Specifically, we constructed CIFAR-10 test minibatches with a 50-50 split of uncorrupted and fully corrupted (intensity level 1-5) data. Thus, the full combined test set, consisted of 20,000 examples. We trained SGD and deep ensembles using code from Izmailov et al. (2021b), and used HMC samples from Izmailov et al. (2021b). The results are presented in Fig. 11. Test-time batchnorm still gives an improvement, and ShiftMatch gives a further improvement for all methods tested (SGD, deep ensembles and HMC), except for SGD for the log-likelihood. Of course, the improvement is less spectacular than when the test minibatches all consist of corrupted data (Fig. 4 and Table. 2).

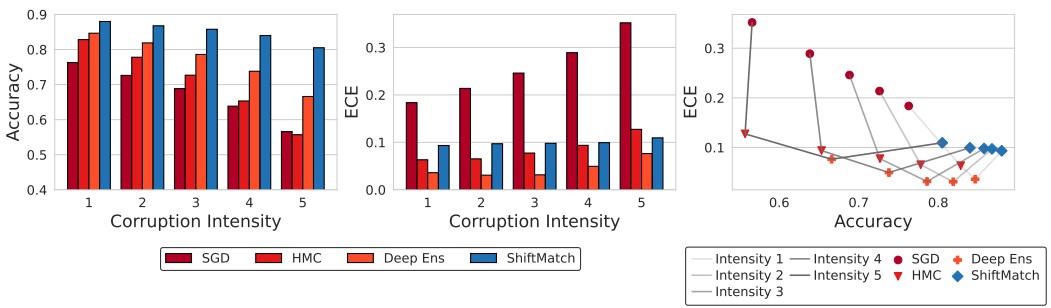

Figure 10: Expected calibration error (ECE) of different methods. We begin by showing the accuracy (left) and ECE (middle) separately, with corruption intensity on the x-axis, for SGD, HMC, deep ensembles and ShiftMatch. Note that, for all experiments, the ECE numbers are computed using the same code and setting. The results for SGD is acquired by averaging 5 independent trials of different random seeds as this method had very high variance in the ECE. The trials are conducted using the code of Izmailov et al. (2021b), which used filter response normalization (FRN) (Singh & Krishnan, 2020) rather than batch normalization (Ioffe & Szegedy, 2015) and did not apply data augmentation. The right most figure directly compares ECE and accuracy under different corruption intensities. Results under the same corruption intensity are connected via grey lines, and different methods are presented with different markers and colours. Note the accuracy of SGD is lower than the numbers reported in Figure. 1, which are directly extracted from Fig. 7 in Izmailov et al. (2021b). We believe that this is due to Izmailov et al. (2021b) using only one random seed, whereas here we averaged over five random seeds trained using their code.

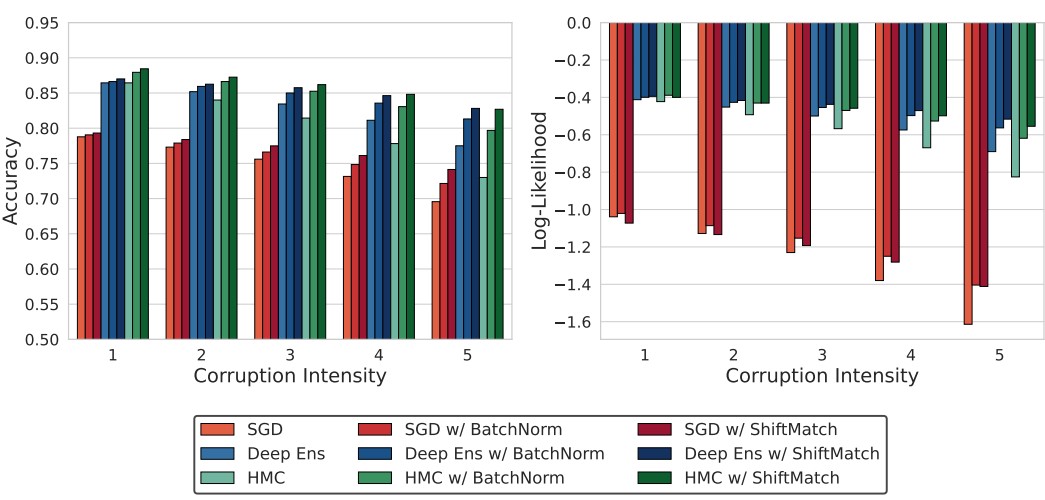

Figure 11: Performance of plain, test-time batchnorm and ShiftMatch in combination with SGD, deep ensembles and HMC on CIFAR-10 where we test on combined minibatches with a 50-50 split of uncorrupted and fully corrupted (intensity level 1-5) data.

