# OpenReview forum: "Robustness to corruption in pre-trained Bayesian neural networks"
_ICLR.cc/2023/Conference — ICLR 2023 poster_

### Official Review · Reviewer_NoHk · 2022-10-20

**Confidence:** 4
**Correctness:** 3
**Technical Novelty And Significance:** 4
**Empirical Novelty And Significance:** Not applicable
**Recommendation:** 8

**Clarity, Quality, Novelty And Reproducibility:**

**Clarity**: The paper is mostly well written and very easy to follow. Some minor comments are listed below.

**Quality**: The paper appears to be technically sound. The method is theoretically justified and rather extensively tested empirically with the exception that recent other methods dealing with covariate shift are missing for comparison.

**Novelty**: The ShiftMatch method appears to be novel.

**Reproducibility**: It seems some small details might be missing for reproducibility. For example, learning rate for SGD in the experiments is not specified, or how the outputs for deep ensembles were averaged.

Other comments/suggestions (these don’t impact the score):

1. Acronyms such as SGD and FRN and HMC are not defined in the text

2. Introduction, first paragraph. “images” – the text prior to that was not limited to images only

3. Figure 1 caption. “HMC, SGD, Deep Ensemble, SGLD, mean-field VI” – missing references

4. After eq.(3): “C is the number of features, … from the inputs to a single feature in the first hidden layer”. If I understand correctly C is the number of features in the input layer, which should be emphasised to distinguish from that single feature in the first hidden layer

5. Page 3, last paragraph. “including for robustness” – weird wording

6. Page 3, last paragraph. “C is the number of channels” – first, channels were not properly introduced here and C has been used before to denote the number of features

7. Around eq. (12)-(13). It is probably worth mentioning that priors are assumed to be the same in the “standard BNN” and in the BNN with ShiftMatch

8. Eq. (14). C has been used already to denote the number of features and the number of channels

9. Figure 3 caption. Ensembles are not mentioned

10. Page 7, bottom. It would be interesting to see a discussion on why on some corruptions the ensemble works better and on some worse than ShiftMatch.

11. Figure 5 caption. ImageNet-C is missing a reference.

12. Appendix B. Which data has been used here? “Tab.3 gives performance differences with and without ShiftMatch” – was it meant to be “batchnorm” here?

Minor:

1. Eq. (10) “.” -> “,”

2. Sec. 4.6. First sentence. “ShiftMatch retains all useful the theoretical properties” -> “ShiftMatch retains all the useful theoretical properties”

3. Conclusion. Second sentence. “Shiftmatch” -> “ShiftMatch”

4. Appendix A. last line. S -> Sigma


**Strength And Weaknesses:**

**update after rebuttal** I have read the authors’ response and other reviews. The authors did clarify mostly my concerns so I am increasing the score.
=======================



Strengths:
* The paper proposes an interesting way to deal with test time data corruptions without the need to touch the training time. This allows a user to use any of the pretrained networks and apply ShiftMatch only at test time. With the caveat that one would need an access to training data to compute statistics required for transformation.
* This transformation is neatly incorporated into Bayesian formulation.
* The paper provides rather extensive empirical evaluation on the method, though check the weaknesses on experiments.
* The paper is well written mostly and easy to follow.

Weaknesses:
* Experiments are missing comparison with other methods that deal with corruptions or covariate shift. The only baseline that directly addresses the same problem is EmpCov, but there are other works, e.g., Trinh et al. 2022. Tackling covariate shift with node-based Bayesian neural networks. ICML.
* (not major) I am missing a bit more discussion how ShiftMatch fits into the Bayesian deep learning. For example, does it help/hurt with cold posterior effect? (Wenzel et al., 2020. How Good is the Bayes Posterior in Deep Neural Networks Really? ICML). Also, there is a discussion on effect (non-decremental as one might think) on OOD detection, but it would be great to see more discussion and experiments on effect of ShiftMatch on uncertainty quantification provided by BNNs.
* (not major) A bit more discussion and experiments on approximations for covariance estimates would be beneficial. I find results from Appendix C are rather surprising and interesting, but lacking more explanation here. It is unclear for me why ““w/I Kronecker factored covariance” compromises the ability to obtain accurate covariance estimates.
* Test time runtime discussion is missing. The paper rightly so emphasises that training time is unaffected by ShiftMatch, but how slower does the test time become with ShiftMatch? There is a brief mentioning of runtime for spatial batchnorm (a potential implementation of ShiftMatch).
* (not major) It would be interesting to see the comparison not only to batchnorm but with decorrelated batchnorm cited in the paper.


**Summary Of The Paper:**

The paper proposes ShiftMatch – a transformation of the training data that represents likelihood in Bayesian neural networks. This data-dependent likelihood helps dealing with corruptions of the data at test time. It can perfectly remove certain types of corruptions. Interestingly, it does not change the training data, so this transformation can be applied on pre-trained networks. Moreover, it can be applied to normal neural networks, not only Bayesian neural networks.

**Summary Of The Review:**

I suggest acceptance of the paper as I believe the strengths outweigh the weaknesses. The only real big weakness is the absence of comparison to other methods other than EmpCov dealing with covariate shift.

I would be interested to know the authors thoughts on the two weaknesses I mentioned: absences of other baselines and runtime discussion – during the rebuttal

---

> ### Author Response · Authors · 2022-11-09
> **Response (2/2)**
>
> > Test time runtime discussion is missing. The paper rightly so emphasises that training time is unaffected by ShiftMatch, but how slower does the test time become with ShiftMatch? There is a brief mentioning of runtime for spatial batchnorm (a potential implementation of ShiftMatch).
>
> For the CIFAR-10 experiments, given a test batch of 10,000 samples, it takes 1.15s for one forward propagation without ShiftMatch and 1.84s with ShiftMatch.  Given the large potential improvements in accuracy, this cost is likely to be worthwhile in many application domains.  We have added this note to Appendix C.
>
> ### Other comments/suggestions (these don’t impact the score):
>
> > Acronyms such as SGD and FRN and HMC are not defined in the text
>
> Fixed
>
> > Introduction, first paragraph. "images" – the text prior to that was not limited to images only
>
> Fixed
>
> > Figure 1 caption. "HMC, SGD, Deep Ensemble, SGLD, mean-field VI" - missing references
>
> Fixed
>
> > After eq.(3): "C is the number of features, ... from the inputs to a single feature in the first hidden layer". If I understand correctly C is the number of features in the input layer, which should be emphasised to distinguish from that single feature in the first hidden layer
>
> Fixed (we now use $M$ for the number of features, and $C$ for the number of channels).
>
> > Page 3, last paragraph. “including for robustness” – weird wording
>
> Fixed
>
> > Page 3, last paragraph. “C is the number of channels” – first, channels were not properly introduced here and C has been used before to denote the number of features
>
> We now use $M$ for the number of features, and introduced channel/height/width first
>
> > Around eq. (12)-(13). It is probably worth mentioning that priors are assumed to be the same in the “standard BNN” and in the BNN with ShiftMatch
>
> fixed
>
> > Eq. (14). C has been used already to denote the number of features and the number of channels
>
> Fixed: changed the matrix $C$ to the matrix $S$.
>
> > Figure 3 caption. Ensembles are not mentioned
>
> Fixed.
>
> > Page 7, bottom. It would be interesting to see a discussion on why on some corruptions the ensemble works better and on some worse than ShiftMatch.
>
> As expected, ShiftMatch performs well when the corruption changes the "texture" of the image, e.g. in brightness, fog, zigzag or canny edges.  At the same time, there are two examples where ShiftMatch performs particularly badly: scale and translate.  Before going on, it is worth remembering that the usual way to deal with scale and translation corruptions is data-augmentation, which was not used in this training procedure, as we were mirroring Izmailov et al. (2021).  On scale and translate, it seems here that ShiftMatch is doing badly, rather than ensembles is doing well, as for rotate, ensembles is the same as HMC and EmpCov, while for translate, ensembles is slightly worse than the other two.  We would speculate that the reason is that both translate and scale are highly non-stationary (i.e. they cannot be corrected by modifying the power spectrum of the image), so we would not expect ShiftMatch to perform well on these images.  At the same time, we would speculate that these translations are changing the spatial structure of the images, and ShiftMatch's attempts at correcting these spatial shifts turns out to be detrimental.
>
> > Figure 5 caption. ImageNet-C is missing a reference.
>
> Fixed.
>
> > Appendix B. Which data has been used here? “Tab.3 gives performance differences with and without ShiftMatch” – was it meant to be “batchnorm” here?
>
> Yes, fixed.
>
> > Eq. (10) "." -> ","
>
> Fixed
>
> > Sec. 4.6. First sentence. "ShiftMatch retains all useful the theoretical properties" -> "ShiftMatch retains all the useful theoretical properties"
>
> > Conclusion. Second sentence. "Shiftmatch" -> "ShiftMatch"
>
> Fixed.
>
> > Appendix A. last line. S -> Sigma
>
> Fixed.

---

> > ### Comment · Reviewer_NoHk · 2022-11-18
> > **Thank you for your response**
> >
> > Thank you for adding the test running time and fixing minor things

---

> ### Author Response · Authors · 2022-11-09
> **Response (1/2)**
>
> > Experiments are missing comparison with other methods that deal with corruptions or covariate shift. The only baseline that directly addresses the same problem is EmpCov, but there are other works, e.g., Trinh et al. 2022. Tackling covariate shift with node-based Bayesian neural networks. ICML.
>
> Thanks for this reference!  We have added it to the related work, and we will investigate adding this method to our plots. As far as we can see, their performance is quite a bit worse than our results.  For instance, in our Fig. 1, we have an accuracy of just over 80\% for ShiftMatch for the highest intensity corruptions on CIFAR-10-C.  In contrast, in Trinh et al. Fig. 10, they have accuracies going up to only around 65\% for the highest intensity corruptions. Further, our Fig. 1 shows big differences from ShiftMatch to strong baselines such as plain deep ensembles.  In contrast, Trinh et al. seems to show only small differences (at least in accuracy) between their method and their baselines.
>
> In general, finding other comparable baselines is quite hard, as we are in a very restrictive setting:
> * our approach does not modify network training (Trinh et al. does propose a new variational objective and requires training a BNN from scratch).
> * our approach is cheap at test-time, and is therefore applicable to large ensembles (unlike e.g. TENT [1] that requires some gradient steps at test-time).
> * ideally, baselines would be in the Bayesian setting with a valid likelihood
>
> But of course we are open to any other suggestions!
>
> [1] Wang, Dequan, et al. "Tent: Fully Test-Time Adaptation by Entropy Minimization." International Conference on Learning Representations. 2020.

---

### Official Review · Reviewer_BEej · 2022-10-25

**Confidence:** 3
**Clarity, Quality, Novelty And Reproducibility:** See the strengths and weaknesses writ…
**Correctness:** 2
**Technical Novelty And Significance:** 3
**Empirical Novelty And Significance:** 3
**Recommendation:** 6

**Strength And Weaknesses:**

Strengths :

- It's ingenious to try to see a change in the prior as a change in the likelihood.

- The performance is good compared to the comparison methods (BatchNorm, EmpCov).

Weaknesses :

- Chapter 4.1, which is the motivation of this paper, is hard to follow. More specific, authors claim that

$w \sim N(0, \frac{1}{C N_{\text{train}}}( X_{\text{train}}^{\top} X_{\text{train}}) )$,

$X_{\text{test}} w \sim N(0, \frac{1}{C N_{\text{train}}} X_{\text{test}} ( X_{\text{train}}^{\top} X_{\text{train}}) X_{\text{test}}^{\top})$

can be seen as

$\operatorname{ShiftEmpCov}(X_{\text{test}} ; X_{\text{train}}) = X_{\text{test}} ( X_{\text{train}}^{\top} X_{\text{train}})^{1/2}$,

$w^{\prime} \sim N(0, I)$,

$\operatorname{ShiftEmpCov}(X_{\text{test}} ; X_{\text{train}})
w^{\prime} \sim N(0, \frac{1}{C N_{\text{train}}} X_{\text{test}} ( X_{\text{train}}^{\top} X_{\text{train}}) X_{\text{test}}^{\top} )$.

This statement is obviously true, but however, I think these statement is related to prior distribution of $X_{\text{test}} w$.
If we change prior like above, then what can you say about the posterior and predictive distribution? I think more detailed explanation is needed.

- Proposed method can be summarized as follows : Match the first and second moments of the test data distribution with the those of the train data distribution. In other words, the algorithm compares test dataset with train dataset, and make test dataset distribution similar to train dataset distribution by covariate shift. I wonder if it will work well even if half of images are corruption-free and the other half are corrupted by intensity 5. Considering the idea of proposed method, I'm worried that it works well only for one unified corruption intensity.

- Is it much more effective to match the distribution in all layers than to match only the distribution of input data? I wonder what it means to consider all the layers.

- It would have been nice to measure the ‘Expected Calibration Error’, which is commonly used to evaluate Bayesian models on classification problem.



**Summary Of The Paper:**

This paper proposes ShiftMatch, a new training-data-dependent likelihood for robustness to corruption in  pre-trained Bayesian neural network. ShiftMatch allows it to use publicly available samples from pre-trained BNNs (Although it needs train data on test phase).

**Summary Of The Review:**

While the subject of this paper is interesting and novel, I believe the authors' claims are not adequately supported due to a series of weaknesses. I am more than willing to increase my score if the authors address the aforementioned limitations.

---

> ### Author Response · Authors · 2022-11-09
> **Response (2/2)**
>
>
> > I wonder if it will work well even if half of images are corruption-free and the other half are corrupted by intensity 5. Considering the idea of proposed method, I'm worried that it works well only for one unified corruption intensity.
>
> Thanks for the suggestion! We did just this, and have added the results to Appendix I. For all methods tested (SGD, deep ensembles and HMC), we still see improvements for ShiftMatch over test-time batchnorm and the plain method, except for the SGD log-likelihood. These improvements are considerable, but are less spectacular than those in Fig. 4.
>
> > Is it much more effective to match the distribution in all layers than to match only the distribution of input data? I wonder what it means to consider all the layers.
>
> We did these ablations in Appendix D in the original manuscript.  In particular, see Figure 7, where the second column "ShiftMatch (input only)" is ShiftMatch applied only to the input layer.  We see that applying ShiftMatch at all layers roughly doubles the benefit of applying ShiftMatch only at the input layers, and that this grows at higher intensities.
>
> > It would have been nice to measure the "Expected Calibration Error", which is commonly used to evaluate Bayesian models on classification problem.
>
> We have added these results in Appendix H, Fig 10. Broadly, SGD has very poor accuracy and ECE, while ShiftMatch performs best in terms of accuracy, and deep ensembles performs best in terms of ECE. We could understand these results as ShiftMatch trading a bit of ECE for a lot of accuracy (see Fig. 10 right). However, comparing ECE is difficult when accuracy differs so dramatically (e.g. for corruption intensity 5) (see [1] for further details). As one example, remember that ECE bins the models confidence (e.g. in the range 0.5--0.6), and asks whether the model's probability of being correct is close to the confidence (in this case, the probability of being correct should be about 0.55). When the model is more accurate, more of its probabilities will be close to 0 or close to 1, meaning that there will be fewer samples in the intermediate bins (e.g. see Figure 3 in [2]). Having fewer samples in the intermediate bins implies that there will be larger errors in the calibration curves just due to finite samples, and larger errors in the calibration curve of course means a higher ECE. This is one example where changes in the ECE may occur not due to changes in the model's actual calibration, but due to finite sampling errors.
>
> In addition, we have results on uncertainty-based OOD detection in Table 1, showing that ShiftMatch does not seem to have a detrimental effect on OOD detection.
>
>
> [1] Nixon J, Dusenberry MW, Zhang L, Jerfel G, Tran D. Measuring Calibration in Deep Learning. CVPR Workshops (2019).
>
> [2] Garriga-Alonso, A., Rasmussen, C.E. and Aitchison, L., 2018. Deep convolutional networks as shallow gaussian processes. ICLR (2019).

---

> > ### Comment · Reviewer_BEej · 2022-11-18
> > **Thank you for your detail explanation**
> >
> > I would like to thank the authors for the efforts put in their responses.
> >
> > These addresse my all concern. Thus, I am increasing my score from 5 to 6.

---

> ### Author Response · Authors · 2022-11-09
> **Response (1/2)**
>
> The key claim in Sec. 4.1 is that models with EmpCov priors and ShiftEmpCov likelihoods have equivalent behaviour, because the distribution of the outputs of a layer with an EmpCov prior and a ShiftEmpCov likelihood are the same.
>
> To formalise this notion of equivalence, we need to look at a function-space viewpoint.   This viewpoint works with the distribution over the logits (or neural network outputs) $L$.  The logits, $L$, depend on the inputs, $X$ and the neural network weights, $W$.  Of course, $L$ is usually a deterministic function of $X$ and $W$.  The function space viewpoint (e.g. [1]) induces a distribution over functions (i.e. logits or network outputs, $L$) by marginalising the prior over weights,
> \begin{align}
>   P(L| X) = \int dW P(L| W, X) P(X).
> \end{align}
> While explicitly computing P(L|X) is usually intractable, this viewpoint is very useful for establishing theoretical properties.  In particular, as the class labels depend only on the logits/network outputs, we can write the full function-space graphical model as,
> \begin{align}
>   X \rightarrow L \rightarrow y.
> \end{align}
> Here, we combine the training and test points, $X=(X_{\rm train}, X_{\rm test})$, $L=(L_{\rm train}, L_{\rm test})$ and $y=(y_{\rm train}, y_{\rm test})$.
> The prior over logits, $P(L| X)$ is the same in both EmpCov and ShiftEmpCov viewpoints, because the prior distribution of the outputs of each layer is the same.
> Likewise, the softmax likelihood $P(y| L)$ is again the same in both EmpCov and ShiftEmpCov viewpoints.
> As the priors are exactly equivalent under the two viewpoints, the posteriors over all quantities here must also be equivalent. We can go further and explicitly write out the equivalent posteriors.
> In particular, we want the posterior over train and test logits, $L$, conditioned on training labels, $y_{\rm train}$,
> \begin{align}
>   P(L| X, y_{\rm train}) &\propto P(y_{\rm train}| L_{\rm train}) P(L|X).
> \end{align}
> Remember that $L = (L_{\rm train}, L_{\rm test})$ and $X=(X_{\rm train}, X_{\rm test})$.
> Again, this posterior is the same for the EmpCov and ShiftEmpCov viewpoints, because the prior, $P(L|X)$, is the same, and because the softmax likelihood, $P(y_{\rm train}| L_{\rm train})$ is the same.
> From the posterior over logits, we can obtain the predictive distribution over test labels,
> \begin{align}
>   P(y_{\rm test}| X, y_{\rm train}) = \int dL P(y_{\rm test}| L) P(L|X, y_{\rm train})
> \end{align}
> and this is again equivalent because the posterior over logits is equivalent, and the softmax likelihood is equivalent.  We have included this in a new Appendix (Appendix B).
>
> [1] Rudner TG, Chen Z, Teh YW, Gal Y. Tractable function-space variational inference in Bayesian neural networks. InICML 2021 Workshop on Uncertainty and Robustness in Deep Learning 2021.

---

### Official Review · Reviewer_MwDb · 2022-10-25

**Confidence:** 4
**Correctness:** 3
**Technical Novelty And Significance:** 3
**Empirical Novelty And Significance:** 3
**Recommendation:** 8

**Clarity, Quality, Novelty And Reproducibility:**

- Clarity and quality: the paper is generally well-written. Motivations, reviews of the existing works, and the design principle of the methods are clearly described. The experimental results are well displayed with easy-to-interpret graphs and figures.
- Novelty: as far as I know, the ShiftMatch likelihood is novel, even though it may have been inspired by some of the existing works.
- Reproducibility: the authors describe details of the experimental settings. Also, all the codes including scripts that can reproduce the results are provided.

**Strength And Weaknesses:**

Strength
- The paper is well-written and easy to follow.
- The problem of interest is indeed an important topic in the literature, so the motivation is clear.
- I find the idea of the data-dependent likelihood that does not alter the training procedure interesting and novel. I think the design of the ShiftMatch likelihood and its interpretation as a data-dependent likelihood is clever.
- The experiments are thorough, and the results are convincing. Including the additional experiments in the appendix, the authors provide various ablations and alternatives that can further strengthen the arguments made in the paper.

Weakness
- If I have to be picky, the method can be considered incremental since it is largely inspired by existing methods including EmpCov and test-time batchnorm. Still, I think the proposed method is valuable, especially the design of ShiftMatch likelihood leaving the training procedure unchanged.
- As already discussed in the paper, the method requires storing or recovering training data for prediction. However, the arguments in section 4.5 to defend against this is also reasonable, so I won't think this is a critical problem.

Here are some questions and minor comments:
- EmpCov seems to transform only the input layers, but ShiftMatch seems to be applied for all the intermediate layers. Can you comment on this difference? For instance, would the performance of ShiftMatch be degraded if only applied to the input layers?
- In page 7, the paper explains the reason why ShiftMatch performs poorly for "Translate" corruption. It seems that ShiftMatch also underperforms baselines for "Scale", and slightly underperforms deep ensembles for "Gauss blur". The latter is quite disappointing since section 4.6 explains how ShiftMatch can "perfectly" removes stationary linear corruptions. Can you comment on this?
- I don't expect the size of the training data to stably estimate the statistics to be large. Having said that, one could think of a bagging-like procedure where multiple subsets of training data are sampled, corresponding statistics are computed, and multiple predictions are computed from the multiple statistics to be ensembled. This may make the interpretation of the data-dependent likelihood more cumbersome, but may enhance the robustness of the method.


**Summary Of The Paper:**

This paper presents ShiftMatch, a simple method based on a data-dependent likelihood, enhancing the robustness of Bayesian neural networks to corrupted data. Given a (pre)trained BNN, ShiftMatch computes the training data statistics (mean, variances) and uses them to construct a transformation matrix for each layer. The prediction for a test input is then done with the transformation matrix computed from the training statistics. The prediction with the training data-dependent transformation alters the likelihood, but this transformation leaves the likelihood evaluated with the training data invariant; that is, even though the likelihood was changed, it remains the same for the training data, so one can just reuse pretrained BNNs trained with typical data-independent categorical likelihoods. For high-dimensional data with spatial structures, the authors further propose structural covariance estimation schemes that could improve the scalability of the method. The benefit of the proposed method is demonstrated via various image classification benchmarks with corruptions.

**Summary Of The Review:**

In summary, I enjoyed reading the paper, and find it to be a novel contribution with promising results.

post-rebuttal: I have read the authors' response which resolved my concerns. I keep my original score.

---

> ### Author Response · Authors · 2022-11-09
> **Response**
>
> Thanks for your positive and careful review!
>
> > EmpCov seems to transform only the input layers, but ShiftMatch seems to be applied for all the intermediate layers. Can you comment on this difference? For instance, would the performance of ShiftMatch be degraded if only applied to the input layers?
>
> We did these ablations in Appendix D in the original manuscript.  In particular, see Figure 7, where the second column "ShiftMatch (input only)" is ShiftMatch applied only to the input layer.  We see that applying ShiftMatch at all layers roughly doubles the benefit of applying ShiftMatch only at the input layers, and that this grows at higher intensities.
>
> Note that applying EmpCov priors at multiple layers is very difficult, because the prior over weights at an intermediate layer depends on the inputs to that layer, and those inputs in turn depend on earlier weights.  This difficulty is likely why they did not use multi-layer EmpCov priors in Izmailov et al. (2021a).
>
> > In page 7, the paper explains the reason why ShiftMatch performs poorly for "Translate" corruption. It seems that ShiftMatch also underperforms baselines for "Scale", and slightly underperforms deep ensembles for "Gauss blur". The latter is quite disappointing since section 4.6 explains how ShiftMatch can "perfectly" removes stationary linear corruptions. Can you comment on this?
>
> In Fig. 3 on page 7, it is "glass blur", not "Gauss blur".  Looking at the original paper (Mu & Gilmer 2019) these are indeed very different from a Gaussian blur.
>
> > I don't expect the size of the training data to stably estimate the statistics to be large. Having said that, one could think of a bagging-like procedure where multiple subsets of training data are sampled, corresponding statistics are computed, and multiple predictions are computed from the multiple statistics to be ensembled. This may make the interpretation of the data-dependent likelihood more cumbersome, but may enhance the robustness of the method.
>
> Agreed!  We will investigate these possibilities in future work!

---

> > ### Comment · Reviewer_MwDb · 2022-11-29
> > **Response to response**
> >
> > Thank you for your response which resolved most of my concerns. I keep my score.

---

### Decision · Program_Chairs · 2023-01-20

**Decision:**

Accept: poster

**Justification For Why Not Higher Score:**

Novelty is fairly limited given related prior work on leveraging training data statistics for robustness. Empirical results could be substantiated as well with more comparison.

**Justification For Why Not Lower Score:**

Reviewers generally found the work well-written and that the experiments were fairly convincing.

**Metareview: Summary, Strengths And Weaknesses:**

This work proposes ShiftMatch, which is a new likelihood motivated to improve the robustness of Bayesian neural networks on corrupted data. The idea is to not change the neural network's likelihood during training, enabling it to use existing, say, pretrained Hamiltonian Monte Carlo samples. Leveraging training data statistics, ShiftMatch then applies the statistics to transform the likelihood at test time.

Reviewers generally found the work well-written and that the experiments were fairly convincing.

Overall, everyone seems quite positive and leans towards accept. I agree with this consensus.

**Note From Pc:**

if the above contains the word "oral" or "spotlight" please see: "oral" presentation means -> notable-top-5% and "spotlight" means -> notable-top-25%. As stated in our emails, we are disassociating presentation type from AC recommendations